# On Local Limits of Sparse Random Graphs: Color Convergence and the Refined Configuration Model

**Alexander Pluska    Sagar Malhotra**
Department of Computer Science
TU Wien
Vienna, Austria

## Abstract

Local convergence has emerged as a fundamental tool for analyzing sparse random graph models. We introduce a new notion of local convergence, *color convergence*, based on the Weisfeiler–Leman algorithm. Color convergence fully characterizes the class of random graphs that are well-behaved in the limit for message-passing graph neural networks. Building on this, we propose the *Refined Configuration Model* (RCM), a random graph model that generalizes the configuration model. The RCM is universal with respect to local convergence among locally tree-like random graph models, including Erdős–Rényi, stochastic block and configuration models. Finally, this framework enables a complete characterization of the random trees that arise as local limits of such graphs.

## 1   Introduction

Understanding the local structure of random graphs is important for analyzing algorithms in network science and machine learning (Van Der Hofstad, 2024; Lovász, 2012; Borgs et al., 2019, 2021; Gamarnik & Sudan, 2014). *Local convergence* characterizes the structure around a randomly chosen node in the large graph limit (Benjamini & Schramm, 2011). However, this level of detail exceeds what can be exploited by color refinement (Weisfeiler & Leman, 1968; Babai et al., 1980; Huang & Villar, 2021a) — a prominent algorithm that bounds the expressivity of Message Passing Graph Neural Networks (MPNNs) (Gori et al., 2005; Scarselli et al., 2009; Xu et al., 2019; Morris et al., 2019). In this paper we introduce a novel notion of local convergence, namely *color convergence*, that is closely aligned with color refinement, and facilitates the analysis of MPNNs through the framework of local convergence. We show that color convergence completely characterizes a general notion of learnability for MPNNs. We then show that the class of color convergent random graphs subsumes widely investigated locally tree-like models (Van Der Hofstad, 2014, 2024) including sparse variants of Erdős–Rényi, stochastic block (Abbe et al., 2014), and configuration models (Bollobás, 1980). Finally, we devise a random graph model, called the *refined configuration model* (RCM), that is universal with respect to color convergence.

**Related Work**

Our work makes key contributions to the areas of graph limits, generalization analysis of MPNNs, and random graph models. In the following, we briefly discuss the related work in these areas.

**Graph limits.** Our work expands the general investigation of random graph limits (Lovász, 2012). Limits for dense random graphs are charecterized by *graphons* (Lovasz & Szegedy, 2004). In the sparse setting, local convergance has emerged as an important tool for analyzing the limit behavior of random graphs (Benjamini & Schramm, 2011; Aldous & Steele, 2004; Aldous & Lyons, 2007; Hatami et al., 2014; Milewska et al., 2025; Dort & Jacob, 2024). In this paper, we introduce a novel

39th Conference on Neural Information Processing Systems (NeurIPS 2025).

relaxation of local convergence, that characterizes the structure around a randomly chosen node in terms of the color refinement algorithm (Weisfeiler & Leman, 1968; Huang & Villar, 2021a).

**MPNN Generalization.** Our work advances the analysis of generalization behavior of MPNNs. Previous works have investigated generalization behavior of MPNNs, e.g., via VC dimension, Rademacher complexity or PAC-Bayesian analysis (Liao et al., 2021; Oono & Suzuki, 2020; Maskey et al., 2022; Li et al., 2022; Tang & Liu, 2023; Rauchwerger et al., 2025; Morris et al., 2023). Most of these works fix a class of random graphs as data distribution and are largely restricted to the dense graph setting. We especially focus on generalization behavior of MPNNs for the node classification task on sparse random graphs (Baranwal et al., 2023). However, instead of making assumptions on the data distributions, we give a complete characterization of random graphs on which MPNNs can be learned. We use a general notion of learnability, which is closely related to the notion of *uniform convergence* in learning theory (Shalev-Shwartz et al., 2010). In particular, we show that MPNNs fail to learn on dense random graphs, extending asymptotic analysis of MPNNs in this setting (Adam-Day et al., 2024; Yehudai et al., 2021).

**Random Graph Models.** All dense random graph models admit a universal limit object (Aldous, 1981; Bloem-Reddy & Teh, 2020; Lovasz & Szegedy, 2004; Lloyd et al., 2012). However, no such universal limit is known in the sparse setting (Lovász, 2012). This has led to a fragmented landscape of sparse random graph models such as the sparse Erdős–Rényi, configuration and preferential attachment model. Despite their differences, a unifying property of many models is their local converge to Galton-Watson trees (GWT) (Janson, 2012), a family of random trees arising from branching processes (Harris, 1963). These trees capture the typical local structure around a uniformly chosen node and serve as a canonical, though not universal, object in the study of local limits.

### Contributions

We introduce a generalization of local convergence, color convergence, based on the color refinement algorithm. Color convergence defines sparse random graph limits based on the *color* of a random node after finitely many iterations of the color refinement algorithm. We present a systematic investigation of color convergence. Specifically,

- We show that color convergence completely characterizes the class of random graphs on which MPNNs achieve *probabilistic consistency of empirical risk* for node classification tasks. That is, color convergent random graphs are exactly the random graphs for which the empirical risk converges to the true risk with high probability for all possible MPNNs.
- We introduce the refined configuration model, a generalization of the configuration model. Leveraging color convergence, we show that it captures the limit behavior of MPNNs on random graphs and is universal with respect to local convergence to Galton–Watson trees. In the limit, it subsumes many widely-investigated sparse random graph models, including Erdős–Rényi, stochastic block and configuration models.

## 2  Background

We use $[n]$ to denote the set $\{0, \ldots, n-1\}$ and $\mathbb{1}_A$ to denote the indicator function of a set $A$.

An *undirected multigraph* $G$ consists of a set of nodes $V$ and a set of edges $E$, possibly containing loops and multi-edges. We use $N(v)$ to denote the multiset of neighbors of a node $v$ and $d_v$ to denote its degree. In this paper, graphs are always finite, undirected multigraphs. Sometimes the nodes $v$ are attributed with a feature $x_v$. We always assume that the space of features is countable and discrete.

A *rooted graph* $B$ is a graph with a designated root node. The *depth* of a node $v$ is its distance to the root. The depth of a rooted graph $B$ is defined as the maximum depth of its nodes. We denote by $\mathcal{B}_k$ the set of isomorphism classes of rooted graphs of depth at most $k$. Given a graph $G$ and node $v$, the *ball* $B_k(v)$ is the graph rooted at $v$, spanned by vertices $w$ with $d(v, w) \leq k$. If $f$ is a function defined on nodes and $B$ is a rooted graph with root $r$, then we may write $f(B)$ instead of $f(r)$, e.g., we may write $N(B)$ for the neighbors and $d_B$ for the degree of the root. An isomorphism between rooted graphs is a graph isomorphism that also maps the root of one graph to the root of the other.

A *tree* is a connected acyclic graph. A *rooted tree* is a rooted graph whose underlying graph is a tree. We denote by $\mathcal{T}_k$ the set of isomorphism classes of rooted graphs of depth $k$. We write $T|_k$ to denote

the rooted subtree of $T$ containing only nodes of depth at most $k$. Each node in a rooted tree, except the root node, has a unique *parent node*, which is the only adjacent node that has a smaller depth. The *children* of a node are the nodes adjacent to it with a larger depth. A *leaf node* is a node without children. The sub-tree $T(v)$ *induced* by a node $v$ is the maximal sub-tree of $T$ containing the node $v$ but not its parent.

All probability spaces considered in this paper are countable and discrete. A *probability mass function* (PMF) $\mu$ is a function that assigns a probability to each outcome. A *random element* is a function whose domain is a probability space. A *stochastic process* is a sequence of random elements. A *random PMF* is a random element whose codomain is the set of PMFs over a fixed space. We say that a sequence of random PMFs $\mu_t$ *converges in probability* to a PMF $\mu$ if, for all outcomes $x$ and $\varepsilon > 0$,

$$P(|\mu_t(x) - \mu(x)| \geq \varepsilon) \to 0 \tag{1}$$

as $t \to \infty$. In the context of random graphs the notion of local convergence in probability is more powerful and useful than, for instance, local weak convergence (Van Der Hofstad, 2024, p.58).

## 2.1 Random Graphs

A *random graph* $G_t$ is a stochastic process $\{G_t\}_{t \in \mathbb{N}}$ such that $G_t$ is distributed on graphs on $t$ vertices. This formulation of a random graph as a stochastic process indexed by graph size naturally aligns with the framework of graph limits.

Of special interest to us are the configuration model and the bipartite configuration model. Since we are only interested in the limit, we consider a version with i.i.d. degrees. (Van Der Hofstad, 2014, 7.6)

**Definition 2.1.** Let $\mu$ be a PMF over $\mathbb{N}$ with finite mean. The *configuration model* $\mathsf{CM}_t(\mu)$ is defined on the vertex set $\{v_i\}_{i \in [t]}$ as follows:

- Each vertex $v_i$ is independently assigned a degree $d_i \sim \mu$.

- Each vertex is given $d_i$ stubs. The stubs are paired uniformly at random to form edges, allowing loops and multi-edges, until there are 0 or 1 stub(s) left.

**Definition 2.2.** Let $\mu_L, \mu_R$ be PMFs over $\mathbb{N}$ with finite means. The *bipartite configuration model* $\mathsf{BCM}_t(\mu_L, \mu_R)$ is defined on the vertex set $\{v_i\}_{i \in [t]}$ as follows:

- Partition the nodes into *left nodes* $L$ and *right nodes* $R$ by assigning each node independently with probability $\frac{\mathbb{E}[\mu_R]}{\mathbb{E}[\mu_L] + \mathbb{E}[\mu_R]}$ to $L$ and otherwise to $R$.

- Each vertex $v_i \in U$ is independently assigned a degree $d_i \sim \mu_U$ for $U \in \{L, R\}$.

- Each vertex is given $d_i$ stubs. The stubs in $L$ are matched uniformly at random with stubs in $R$ to form edges, allowing multi-edges, until there are no more stubs left in $L$ or $R$.

## 2.2 Galton-Watson Trees

Another important graph-valued random process, with applications in population genetics, computer science, and beyond, is the Galton–Watson tree. See for instance (Van Der Hofstad, 2024, 3.4).

**Definition 2.3.** A *multi-type Galton–Watson tree* (GWT) $W_t$ is a stochastic process $\{W_t\}_{t \in \mathbb{N}}$, where $W_t$ is a random rooted tree of depth $t$. It is parameterized by:

- a finite or countable set of types $S$, with a type-to-feature mapping $s \mapsto x_s$,

- an initial PMF $\mu_0$ over $S$,

- for each $s \in S$, a PMF $\mu_s$ over MultiSet($S$), the set of finite multisets of types.

The process is defined inductively:

- $W_0$ consists of a root node $r$ with type $s_r \sim \mu_0$ and feature $x_r = x_{s_r}$.

- Given $W_t$, generate $W_{t+1}$ by extending each leaf node $v$ at depth $t$:

- For each node $v$, sample a multiset $\{s_1, \dots, s_n\} \sim \mu_{s_v}$.
- For each type $s_i$ in the multiset, attach a child $w$ to $v$ with type $s_i$ and feature $x_{s_i}$.

**Example 2.4.** Suppose $S = \{\textcolor{red}{\bullet}, \textcolor{blue}{\bullet}\}$ and $x_s = s$. Let $W_t$ be parametrized as follows:

$$\mu_0(s) = \begin{cases} \frac{3}{5} & s = \textcolor{red}{\bullet} \\ \frac{2}{5} & s = \textcolor{blue}{\bullet} \end{cases} \qquad \mu_{\textcolor{red}{\bullet}}(A) = \begin{cases} \frac{2}{3} & A = \{\{\textcolor{red}{\bullet}, \textcolor{blue}{\bullet}\}\} \\ \frac{1}{3} & A = \{\{\textcolor{red}{\bullet}, \textcolor{red}{\bullet}\}\} \\ 0 & \text{otherwise} \end{cases} \qquad \mu_{\textcolor{blue}{\bullet}}(A) = \begin{cases} \frac{1}{2} & A = \{\{\textcolor{red}{\bullet}, \textcolor{red}{\bullet}, \textcolor{blue}{\bullet}\}\} \\ \frac{1}{2} & A = \{\{\textcolor{blue}{\bullet}\}\} \\ 0 & \text{otherwise} \end{cases}$$

Then $W_1$ has the support and probability distribution depicted in figure 1.

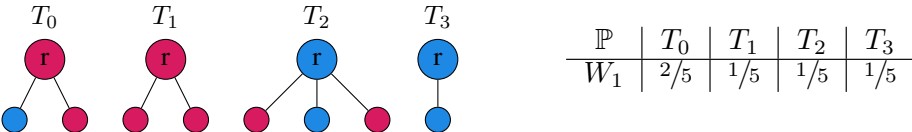

| $\mathbb{P}$ | $T_0$ | $T_1$ | $T_2$ | $T_3$ |
|---|---|---|---|---|
| $W_1$ | $2/5$ | $1/5$ | $1/5$ | $1/5$ |

Figure 1: Support and distribution of $W_1$.

## 2.3 Local Convergence

We focus on local convergence in probability, which is most conveniently defined as the convergence of a sequence of random PMFs (Van Der Hofstad, 2024, Remark 2.13).

**Definition 2.5.** For a random graph $G_t$ and $t \in \mathbb{N}$ we define the random PMF $b_{k,t}$ on $\mathcal{B}_k$ by

$$b_{k,t}(B) = t^{-1} \cdot |\{v \in V(G_t) : B_k(v) \simeq B\}| \,.$$

If $b_{k,t}$ converges in probability as $t \to \infty$, we denote its limit with $b_{k,\infty}$. If $b_{k,\infty}$ is defined, we call $G_t$ $\mathcal{B}_k$-*convergent*. If $G_t$ is $\mathcal{B}_k$-*convergent* for all $k \in \mathbb{N}$ we call $G_t$ *locally convergent*. Furthermore, we call $G_t$ locally tree-like if, for all $\varepsilon > 0$, as $t \to \infty$,

$$\mathbb{P}(t^{-1} \cdot |\{v \in V(G_t) : B_k(v) \text{ contains a cycle}\}| \geq \varepsilon) \to 0.$$

**Example 2.6.** Let $G_t$ be a random graph that, with probability $1/2$, is either a set of $t$ isolated vertices or a cycle on $t$ vertices. Then $G_t$ does not converge locally:

For every $k \in \mathbb{N}$ and $t \geq 2k + 2$, the random PMF $b_{k,t}$ on $\mathcal{B}_k$ is, with probability $1/2$, either $\mathbb{1}_{K_1}$ or $\mathbb{1}_{P_{2k+1}}$, where $K_1$ denotes the singleton graph and $P_{2k+1}$ denotes the path graph of length $2k + 1$. That is, for any PMF $\mu$ on $\mathcal{B}_k$, we have

$$P(|b_{k,t}(K_1) - \mu(K_1)| \geq 1/2) \geq 1/2 \quad \text{or} \quad P(|b_{k,t}(P_{2k+1}) - \mu(P_{2k+1})| \geq 1/2) \geq 1/2.$$

**Theorem 2.7** (Van Der Hofstad (2024)). *Let $G_t = \mathsf{CM}_t(\mu)$. Then $G_t$ converges locally to a GWT. That is, there exists a GWT $W_t$ such that for all $k \in \mathbb{N}$ and $B \in \mathcal{B}_k$ $b_{k,\infty}(B) = \mathbb{P}(W_k \simeq B)$.*

*Remark* 2.8. Many important families of random graphs converge locally to Galton-Watson trees such as inhomogeneous random graphs, including sparse version of Erdős–Rényi and stochastic block models. (Van Der Hofstad, 2024, 3.14)

A central property of any locally convergent random graph is unimodularity (Aldous & Lyons, 2007), or, equivalently *involution invariance* (Hatami et al., 2014). Intuitively, in captures the idea that, statistically, the local limit looks the same from the root as from anywhere else. Whether unimodularity of a limit object is sufficient for the existence of a random graph with said limit is open in general (Aldous & Lyons, 2007, 10.1). For certain classes of limit objects, including versions of Galton-Watson trees, this implication is known (Aldous & Lyons, 2007; Benjamini et al., 2015).

## 2.4 Message Passing Graph Neural Networks

*Message passing graph neural networks* (MPNNs) are a class of deep learning models that operate on graphs. MPNNs iteratively combine the feature vector of every node with the multiset of feature vectors of its neighbors. Formally, let $\mathsf{agg}_k$ and $\mathsf{comb}_k$ for $k \in [l]$ be *aggregation* and *combination*

functions. We assume that each node has an associated initial feature vector $x_v = x_v^{(0)}$. An MPNN $f$ computes a vector $x_v^{(k)}$ for every node $v$ via the following recursive formula

$$x_v^{(k+1)} = \mathsf{comb}_{k+1}(x_v^{(k)}, \mathsf{agg}_{k+1}(\{\!\{x_w^{(k)} : w \in N(v)\}\!\})), \tag{2}$$

where $k \in [l]$. We call $f(v) := x_v^{(l)}$ the *output* of the MPNN. Note that, in our setting, MPNNs do not include a global readout mechanism. Equation (2) subsumes classical MPNN architectures such as GraphSAGE (Hamilton et al., 2017), GCN (Kipf & Welling, 2017), and GIN (Xu et al., 2019), as well as more abstract valuation mechanisms.

## 2.5 Color Refinement

The color refinement algorithm (Weisfeiler & Leman, 1968; Huang & Villar, 2021b) is an important component of modern graph isomorphism test procedures. Starting with an initial coloring, the algorithm repeatedly updates the color of each vertex by aggregating the colors of its neighbors.

**Definition 2.9.** Given a graph $G$ we inductively define a sequence of rooted-tree-valued vertex colorings $\mathsf{cr}_k$:

- $\mathsf{cr}_0(v)$ simply consists of a single root node with feature vector $x_v$.

- $\mathsf{cr}_{k+1}(v)$ comprises a root node $r$ with feature vector $x_r = x_v$ and, for each neighbor $w$ of $v$, the tree $\mathsf{cr}_k(w)$ connected to $r$ via its root.

For a given graph $G$, color refinement eventually stabilizes. That is, there exists $k_0 \in \mathbb{N}$ such that

$$\mathsf{cr}_{k_0}(v) = \mathsf{cr}_{k_0}(w) \iff \mathsf{cr}_k(v) = \mathsf{cr}_k(w).$$

for all $k \geq k_0$ and $v, w \in V$. $\mathsf{cr}_{k_0}$ is known as the *stable coloring* of $G$. Isomorphic graphs have the same stable coloring, although the converse does not necessarily hold.

**Example 2.10.** Consider the graph $G$ below. $\mathsf{cr}_3(v_3)$ yields the illustrated rooted tree.

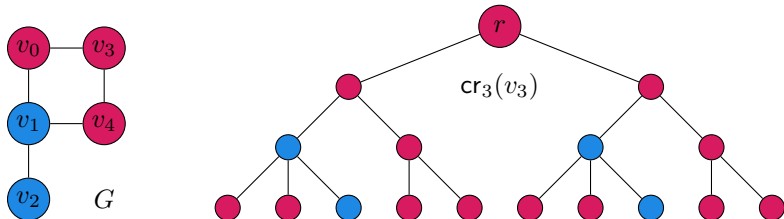

Importantly, color refinement bounds the expressivity of graph neural networks:

**Theorem 2.11** (Xu et al. (2019); Morris et al. (2019)). *For all $v, w \in V$ and $k \in \mathbb{N}$ the following are equivalent:*

- *For all $k$-layer MPNNs $f$ we have $f(v) = f(w)$.*

- $\mathsf{cr}_k(v) = \mathsf{cr}_k(w)$.

## 2.6 Learning and Data Generation Model

Existing work investigates specific parameterizations of sparse random graphs (Baranwal et al., 2023), fixed graph size distributions (Maskey et al., 2022), or restricted classes of MPNNs on dense random graphs (Adam-Day et al., 2024), whereas we aim for a general characterization of learnability for large graphs. We assume that a large graph $G$ with $t$ nodes is sampled from a random graph model $G_t$, followed by a uniform sampling of nodes, on which a node classification task is performed. This setting reflects real-world scenarios such as social, biological, and epidemiological networks, where a *single* large network emerges from an underlying random process, and the MPNNs are used for node classification.

Our goal is to characterize distributions where learning an MPNN from a single, large graph can achieve reliable generalization for node classification tasks. We formalize this by defining *probabilistic consistency of the empirical risk*:

**Definition 2.12.** Let $G_t$ be a random graph and let $f_*$ be an arbitrary node labeling function expressible by a ($k$-layer) MPNN. We say that $G_t$ admits *probabilistic consistency of the empirical risk with respect to ($k$-layer) MPNNs* if the generalization gap goes to zero with high probability in the large graph limit. That is, for every $\varepsilon > 0$ and any ($k$-layer) MPNN $f$, we have that

$$\mathbb{P}\big(|R_{\mathrm{emp}}(f, G_t) - R(f)| \geq \varepsilon\big) \to 0 \quad \text{as } t \to \infty,$$

where the empirical risk and true risk are defined by

$$R_{\mathrm{emp}}(f, G_t) := t^{-1} \cdot |\{v \in G_t : f(v) \neq f_*(v)\}|, \quad R(f) := \lim_{t \to \infty} \mathbb{E}[R_{\mathrm{emp}}(f, G_t)].$$

Notably, all locally convergent random graphs admit probabilistic consistency of empirical risk. However, local convergence is not a necessary condition. In section 3 we define a completely equivalent limit notion, color convergence.

## 3 Color Convergence

We aim to characterize and examine the class of random graphs that satisfy probabilistic consistency of empirical risk with respect to MPNNs. To this end, we reconcile ideas from color refinement and local convergence, and introduce the notion of color convergence. Unlike local convergence, which is defined via distributions over rooted graphs, color convergence is defined via distributions over the set of rooted trees or colors as given in Definition 2.9.

**Definition 3.1.** The set $\mathcal{C}_k \subseteq \mathcal{T}_k$ of *refinement colors* of depth $k$ comprises the isomorphism classes of rooted trees $T \in \mathcal{T}_k$ that occur as the result of color refinement. That is, trees $T$ such that $T \simeq \mathrm{cr}_k(v)$ for some graph $G$ and vertex $v \in V(G)$.

There is a convenient structural characterization of the elements of $\mathcal{C}_k$.

**Proposition 3.2.** *A rooted tree $T \in \mathcal{T}_k$ belongs to $\mathcal{C}_k$ if and only if for every node $v \in V(T)$ that is neither the root nor at depth $k$, there exists a child $c$ of $v$ such that $T(c) \simeq T(p)|_d$, where $p$ is the parent of $v$ and $d$ is the depth of the subtree $T(c)$.*

**Example 3.3.** The tree $T$ belongs to $\mathcal{C}_3$. Every node at depth 1 has a child $c$ such that $T(c) \cong T|_1$, and every node at depth 2 has a child with the same color as its parent. $T'$ does not belong to $\mathcal{C}_3$: the nodes at depth 1 in $T'$ lack a child $c$ satisfying the condition $T(c) \cong T(p)|_d$. Note that $\mathrm{cr}_3(T') = T$.

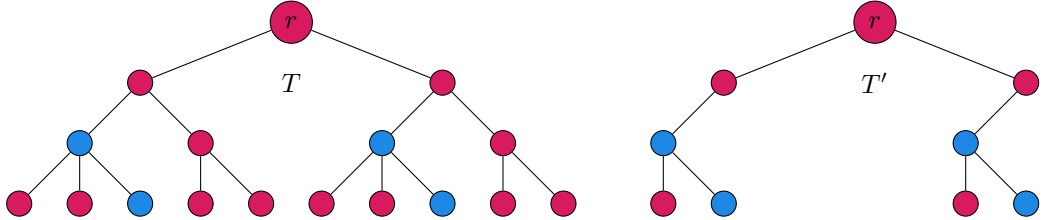

Figure 2: Example demonstrating Proposition 3.2.

We can now give an analogue to local convergence based on refinement colors:

**Definition 3.4.** For any random graph $G_t$ and $t \in \mathbb{N}$ we define the random PMF $c_{k,t}$ on $\mathcal{C}_k$ by

$$c_{k,t}(T) = t^{-1} \cdot |\{v \in V(G_t) : \mathrm{cr}_k(v) \simeq T\}|.$$

If $c_{k,t}$ converges in probability as $t \to \infty$, we denote its limit with $c_{k,\infty}$. If $c_{k,\infty}$ is defined, we call $G_t$ $\mathcal{C}_k$-*convergent*. If $G_t$ is $\mathcal{C}_k$-*convergent* for all $k \in \mathbb{N}$ we call $G_t$ *color convergent*.

### 3.1 Color Convergence and Generalization Gap in MPNNs

In the following example we show that learning MPNNs on random graphs that do not admit color convergence can lead to pathological generalization behavior.

**Example 3.5.** Recall Example 2.6 where $G_t$ is, with probability $1/2$, either a set of $t$ isolated vertices or a cycle on $t$ vertices. Consider the node label $f_*(v) = \mathbb{1}_{\{d_v = 0\}}$, which classifies isolated nodes. Let $f$ denote the constant 0 classifier. Then $\mathbb{P}(R_{\mathrm{emp}}(f, G_t) = 0) = 1/2$ for $t \in \mathbb{N}$ but $R(f) = 1/2$.

Example 3.5 shows that there exists a data distributions (expressed by the random graph $G_t$ in the example) for which an arbitrarily large sample (measured in graph size) can still lead to a constant non-zero generalization gap. In the following theorem we show that such pathological behavior can not occur for color convergent random graphs, i.e., any MPNN we learn on a color convergent random graph generalizes well in the limit with high probability. In fact, we show that for distributions that are not color convergent there is always an MPNN for which the generalization gap does not vanish.

**Theorem 3.6.** *Let $G_t$ be a random graph. The following are equivalent:*

- *$G_t$ is $\mathcal{C}_k$-convergent.*

- *$G_t$ satisfies probabilistic consistency of empirical risk with respect to $k$-layer MPNNs.*

The key insight behind the proof of Theorem 3.6 is that color refinement corresponds to a maximally discriminative MPNN: If a $k$-layer MPNN has a non-zero generalization gap we can identify a distinct refinement color on which this discrepancy arises. In such cases, $c_{k,t}$ cannot converge in probability.

**Corollary 3.7.** *Let $G_t$ be a random graph. The following are equivalent:*

- *$G_t$ is color convergent.*

- *$G_t$ satisfies probabilistic consistency of empirical risk with respect to MPNNs.*

Corollary 3.7 separates random graph models in terms of MPNN learnability. Sparse SBM, preferential attachment models, and inhomogeneous random graphs satisfy color convergence, while dense random graphs (like SBM and Erdős-Rényi) with growing average degree are incompatible with MPNN learning.

**Example 3.8.** Consider a node classifier $f_*$ which expresses the parity of degree, i.e. $f_*(v) = 0$ if $d_v$ is even, and $f_*(v) = 1$ otherwise, and let $G_t$ be an Erdős–Rényi graph with edge probability $p \neq 0$. For $t \in \mathbb{N}$ let $f_t$ denote the following node labeling function:

$$f_t(v) = \begin{cases} f_*(v) & \text{if } d_v < t \\ 1 - f_*(v) & \text{otherwise} \end{cases}$$

By construction, $R_{emp}(f_t, G_t) = 0$ with probability 1 for all $t \in \mathbb{N}$. Since almost all nodes have degree greater than $t$ in the limit, we have $R(f_t) = 1$. This is impossible for color convergent $G_t$.

## 3.2  Properties of Color Convergent Random Graphs

Color convergence is a strict relaxation of local convergence:

**Proposition 3.9.** *Let $G_t$ be a $\mathcal{B}_k$-convergent random graph. Then $G_t$ is $\mathcal{C}_k$-convergent.*

*Remark* 3.10. There are random graphs which are color convergent but not locally convergent: Suppose $G_t$ deterministically, i.e. with probability 1, consists of:

- $t/3$ cycles of length 3, if $t$ is a multiple of 3,

- a single $t$-cylce, otherwise.

$\mathsf{cr}_k(v)$ is the perfect binary tree $BT_k$ of depth $k$ for all $t \in \mathbb{N}$, $v \in G_t$. Therefore, $G_t$ is color convergent. On the other hand, $G_t$ is not $\mathcal{B}_1$-convergent: $b_{1,t}$ is $\mathbb{1}_{C_3}$ when $t$ is a multiple of 3 and $\mathbb{1}_{P_3}$ otherwise, where $C_3$ denotes the 3-cycle and $P_3$ denotes the 3-node path.

In the case of tree-like random graphs, however, both notions are equivalent.

**Proposition 3.11.** *Suppose $G_t$ is a locally tree-like and $\mathcal{C}_k$-convergent. Then $G_t$ is $\mathcal{B}_k$-convergent and, for $T \in \mathcal{T}_k$, we have*

$$b_{k,\infty}(T) = c_{k,\infty}(\mathsf{cr}_k(T)).$$

From a learning perspective, the distribution $c_{k,\infty}$ completely captures the limit behavior of GNNs on a given random graph $G_t$. However, not all distributions on $\mathcal{C}_k$ can arise as such a limit.

**Definition 3.12.** A PMF $\mu$ on $\mathcal{C}_k$ is *sofic* if there exists a random graph $G_t$ such that $\mu = c_{k,\infty}$.

In the case of local convergence, involution invariance is an important property of sofic PMFs. We define an analogous notion for color convergence. It is going to play an important role in the motivation and analysis of our random graph model in Section 4.

**Definition 3.13.** Let $\mu$ be a PMF on $\mathcal{C}_k$ and define

$$d_\mu := \sum_{T \in \mathcal{C}_k} d_T \cdot \mu(T).$$

If $d_\mu < \infty$, we define the *edge-type marginal* $\overline{\mu}$, a PMF on $\mathcal{C}_{k-1}^2$, as

$$\overline{\mu}(T_0, T_1) := \frac{1}{d_\mu} \cdot \sum_{\substack{T \in \mathcal{C}_k \\ T|_{k-1} = T_0}} |\{c \in N(T) : T(c) = T_1\}| \cdot \mu(T).$$

If $d_\mu < \infty$ and $\overline{\mu}$ is symmetric we call $\mu$ *involution invariant with finite degree*.

We can interpret $d_\mu$ as the average degree and $\overline{\mu}$ as a PMF over the pair of vertex colors of a uniformly chosen pair of connected vertices: their edge-type. Symmetry expresses that the probability of a given edge-type $(T_0, T_1)$ equals that of the inverse edge type $(T_1, T_0)$. Analogous to local convergence, our notion of involution invariance with finite degree is a necessary condition for soficity.

**Theorem 3.14.** *Let $k \geq 2$, $\mu$ a sofic PMF on $\mathcal{C}_k$. Then $\mu$ is involution invariant with finite degree.*

Note that $\mathbb{E}[2 \cdot |E(G_t)|] = t \cdot d_{c_{1,t}}$ and $d_{1,t} \to d_{1,\infty} < \infty$ as $t \to \infty$. That is, Theorem 3.14 immediately implies that color convergent random graphs must be sparse.

**Corollary 3.15.** *Let $k \geq 2$ and suppose $G_t$ is $\mathcal{C}_k$-convergent. Then $\mathbb{E}[|E(G_t)|] \in O(t)$.*

## 4 Refined Configuration Model

We are now ready to introduce our generalization of the configuration model. We show its universality with regard to color convergence, local convergence to GWTs and limit behavior of MPNNs.

**Definition 4.1.** The *refined configuration model* $\mathsf{RCM}_t(\mu)$ is parametrized by:

- a finite or countable set of types $S$, with a type-to-feature mapping $s \to x_s$,
- a PMF $\mu$ over $S \times \mathrm{Multiset}(S)$, the product of types and finite multisets of types.

$\mathsf{RCM}_t(\mu)$ is defined on $\{v_i\}_{i \in [t]}$ as follows:

- For each node $v_i$ assign a type-multiset pair $(s_i, A_i) \sim \mu$ independently at random. $s_i$ determines the type of $v_i$, while $A_i$ determines the types of nodes $v_i$ may be connected to. Let $U_s := \{v_i \mid s_i = s\}$ denote the set of nodes which are assigned type $s \in S$.

- For each type $s \in S$, we independently generate a configuration model on $U_s$:
  - Each vertex $v_i$ with $s_i = s$ is given a stub for each occurrence of $s$ in $A_i$. The stubs are paired uniformly at random to form edges, until there are 0 or 1 stubs left.

- For each pair of distinct types $s_L \neq s_R$, we independently generate a bipartite configuration model between $U_{s_L}$ and $U_{s_R}$:
  - Each $v_i \in U_{s_L}$ is given a stub for each occurrence of $s_R$ in $A_i$. Each $v_i \in U_{s_R}$ is given a stub for each occurrence of $s_L$ in $A_i$. Then the stubs in $U_{s_L}$ are matched uniformly at random with the stubs in $U_{s_R}$ to form edges, until there are no more stubs left in $U_{s_L}$ or $U_{s_R}$.

**Example 4.2.** Let $S = X = \{\bullet, \bullet\}$ and $x_s = s$. Consider vertices $\{v_i\}_{i \in [5]}$ and, without specifying the exact distribution, suppose $\mu$ assigns type-multiset pairs as illustrated in Figure 3. The generation process occurs in 3 independent steps:

- Steps 1 & 2: Configuration models are generated on nodes of type $\bullet$ and $\bullet$, respectively.
- Step 3: A bipartite configuration model is generated between nodes of type $\bullet$ and $\bullet$.

The possible results are given as $G_0$ and $G_1$, which occur with probability $2/3$ and $1/3$, respectively.

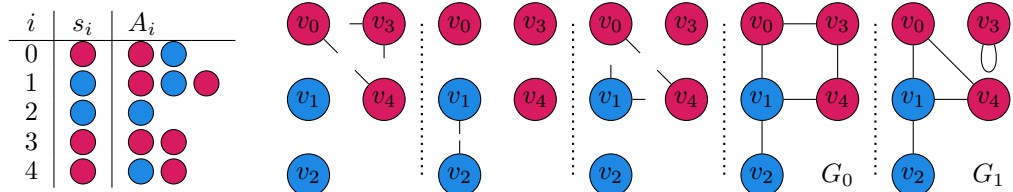

Figure 3: From left to right, we have: a table containing the assigned types, stubs occurring in steps 1, 2, and 3, and possible outputs $G_0$ and $G_1$ of the RCM algorithm.

For general distributions $\mu$, $\mathsf{RCM}_t(\mu)$ may be poorly behaved. For instance, a type $s$ could occur with probability 0 as a node type $s_i$ but multiple times in $A_i$ with high probability. In this case $\mu$ would suggest very different neighborhoods from those observed empirically. We give a condition which guarantees local convergence to the distribution that $\mu$ would lead you to expect.

**Definition 4.3.** Let $\mu$ be a PMF on $S \times \mathsf{MultiSet}(S)$ and define

$$d_\mu := \sum_{s \in S} \sum_{A \in \mathsf{MultiSet}(S)} |A| \cdot \mu(s, A).$$

If $d_\mu < \infty$, we define the *edge-type marginal* $\bar{\mu}$, a PMF on $S^2$, as

$$\bar{\mu}(s_0, s_1) := \frac{1}{d_\mu} \cdot \sum_{A \in \mathsf{MultiSet}(S)} |\{\!\{a \in A : a = s_1\}\!\}| \cdot \mu(s_0, A).$$

If $d_\mu < \infty$ and $\bar{\mu}$ is symmetric we call $\mu$ *involution invariant with finite degree*.

Given $s \in S$, the subgraph of $\mathsf{RCM}_t(\mu)$ spanned by $U_s$ is simply the configuration model $\mathsf{CM}_{|U_s|}(\nu)$, where

$$\nu(n) = \frac{1}{Z_s} \sum_{\substack{A \in \mathsf{MultiSet}(S) \\ \{\!\{s' \in A : s' = s\}\!\} = n}} \mu(s, A) \qquad\qquad Z_s := \sum_{A \in \mathsf{MultiSet}(S)} \mu(s, A)$$

Given distinct types $s_L \neq s_R$, involution invariance with finite degree of $\mu$ ensures that the bipartite subgraph of $\mathsf{RCM}_t(\mu)$ between $U_{s_L}$ and $U_{s_R}$ is the bipartite configuration model $\mathsf{BCM}_{|U_{s_L}|+|U_{s_R}|}(\nu_L, \nu_R)$ with

$$\nu_L(n) = \frac{1}{Z_{s_L}} \sum_{\substack{A \in \mathsf{MultiSet}(S) \\ \{\!\{s' \in A : s' = s_R\}\!\} = n}} \mu(s_L, A) \qquad\qquad \nu_R(n) = \frac{1}{Z_{s_R}} \sum_{\substack{A \in \mathsf{MultiSet}(S) \\ \{\!\{s' \in A : s' = s_L\}\!\} = n}} \mu(s_R, A)$$

It follows that $\mathsf{RCM}_t(\mu)$ converges locally to the following Galton-Watson tree:

**Theorem 4.4.** *Suppose $G_t = \mathsf{RCM}_t(\mu)$ parametrized by types $S_0$, type-to-feature mapping $s \to x_s$ and $\mu$ is involution invariant with finite degree. Then $G_t$ converges locally to the GWT $W_t$ parametrized by type set $S = (\{\bot\} \cup S_0) \times S_0$, type-to-feature mapping $(s_0, s_1) \mapsto x_{s_1}$ and*

$$\mu_0(\bot, s) = Z_s \qquad\qquad \mu_0(s_0, s_1) = 0 \tag{3}$$

$$\mu_{\bot, s}(A) = \begin{cases} \frac{1}{Z_s} \mu(s, \{\!\{q : (p, q) \in A\}\!\}) & \forall (p, q) \in A : p = s \\ 0 & \textit{otherwise} \end{cases} \tag{4}$$

$$\mu_{s_0, s_1}(A) = \frac{|\{\!\{s \in A : s = (s_1, s_0)\}\!\}| \cdot \mu_{\bot, s_1}(A)}{\sum\limits_{B \in \mathit{MultiSet}(S)} |\{\!\{s \in B : s = (s_1, s_0)\}\!\}| \cdot \mu_{\bot, s_1}(B)}. \tag{5}$$

*for all $s_0, s_1 \in S_0$.*

Intuitively, the state-pair $(s_0, s_1)$ of each node is aware of its own state $s_1 \in S_0$, but also its parent's state $s_0 \in S_0$. Root nodes have no parent (3). Given a root $r$, $\mu_{\bot, r}(A)$ is given by $\mu(s, A)$ conditioned on $s = r$, and requiring each element $(p, q)$ of $A$ to come with the correct parent type $p = r$ (4). Finally, $\mu_{s_0, s_1}$ is simply $\mu_{\bot, s_1}$ conditioned on there being a neighbor of type $s_0$ (5).

With this, we can construct a refined configuration model that is color convergent to any sofic PMF $\nu$ on $\mathcal{C}_k$, and obtain universality of the refined configuration model with respect to MPNN learnability.

**Corollary 4.5.** *Suppose $\nu$ is a sofic distribution over $\mathcal{C}_k$. Let $S = \mathcal{C}_{k-1}$ and consider the PMF $\mu$ on $S \times MultiSet(S)$ defined as follows:*

$$\mu(s, A) = \begin{cases} \nu(T) & \text{if there exists } T \in \mathcal{C}_k \text{ such that } T|_{k-1} = s \text{ and } \{\!\!\{ T(c) \,|\, c \in N(T) \}\!\!\} = A \\ 0 & \text{otherwise}. \end{cases}$$

*Then $\mu$ is involution invariant with finite degree and for $G_t = \mathsf{RCM}_t(\mu)$ we have $c_{k,\infty} = \nu$.*

**Corollary 4.6.** *Let $G_t$ be a random graph. The following are equivalent:*

- *$G_t$ satisfies probabilistic consistency of empirical risk with respect to $k$-layer MPNNs*

- *There exists an involution invariant PMF $\mu$ with finite degree such that $\mathsf{RCM}_t(\mu)$ is equivalent to $G_t$ in probability for all $k$-layer MPNNs $f, f_*$. That is, for all $\varepsilon > 0$, as $t \to \infty$,*

$$P(|R_{emp}(f, \mathsf{RCM}_t(\mu)) - R_{emp}(f, G_t)| \geq \varepsilon) \to 0.$$

Note that this holds only for fixed $k$. There may be deep dependencies which our model can not capture. However, for Galton-Watson trees we show that this is not the case:

**Theorem 4.7.** *The following are equivalent:*

- *$W_t$ is the local limit of $\mathsf{RCM}_t(\mu)$ for some involution invariant PMF $\mu$ with finite degree.*

- *$W_t$ is a GWT that arises as the local limit of some random graph.*

*Remark* 4.8. Analogously to the configuration and bipartite configuration model, the refined configuration model can be sampled in time proportional to the expected number of stubs. If $d_\mu < \infty$, we can sample $\mathsf{RCM}_t(\mu)$ in linear time $O(t)$ with high probability.

## 5 Discussion, Limitations and Broader Impact

We have introduced color convergence and shown that it characterizes learnability by MPNNs in the large graph limit. We have investigated connections between color convergence, local convergence, and Galton–Watson trees. We have introduced the refined configuration model — a tractable random graph model which is universally expressive with respect to local limit behavior of MPNNs.

**Limitations and Future Work.** Although our results fill a significant gap in the understanding of MPNNs on sparse random graphs they are inherently limited by the expressivity of color refinement and cannot represent features used by more expressive machine learning architectures, such as higher-order graph neural networks (Maron et al., 2019) or models leveraging subgraph information (Bouritsas et al., 2022). These limitations could, for instance, be addressed by considering graph limits with respect to higher-dimensional versions of the Weisfeiler–Leman algorithm. Furthermore, while we have established a criterion for the learnability of MPNNs, we have not established formal guarantees regarding convergence rates or error bounds. This limits our ability to rigorously quantify the approximation quality or sample complexity of learning in this setting. A natural direction for future work is to investigate under what conditions this framework can yield stronger learnability guarantees. The refined configuration model defines a generative model for structured random graphs. Its practical utility as a data model requires further investigation: Does it admit natural learning algorithms tailored to its structure? Additionally, it is worth exploring whether specific restrictions on its parametrization give rise to interesting subclasses.

**Broader Impact.** Understanding the theoretical foundations of graph neural networks is essential for ensuring their robust and transparent use in high-stakes domains such as drug discovery, recommender systems, and social network analysis. Our framework aims to contribute to the development of principled, interpretable, and responsible machine learning on graphs.

## 6 Acknowledgements

AP acknowledges the support of the project VASSAL: "Verification and Analysis for Safety and Security of Applications in Life" funded by the European Union under Horizon Europe WIDERA Coordination and Support Action/Grant Agreement No. 101160022. SM acknowledges the support of FWF and ANR project NanOX-ML (6728)

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

# A Color Convergence Proofs

**Definition A.1.** The set $\mathcal{C}_k \subseteq \mathcal{T}_k$ of *refinement colors* of depth $k$ comprises the isomorphism classes of rooted trees $T \in \mathcal{T}_k$ that occur as the result of color refinement. That is, trees $T$ such that $T \simeq \mathsf{cr}_k(v)$ for some graph $G$ and vertex $v \in V(G)$.

There is a convenient structural characterization of the elements of $\mathcal{C}_k$.

**Proposition A.2.** *A rooted tree $T \in \mathcal{T}_k$ belongs to $\mathcal{C}_k$ if and only if for every node $v \in V(T)$ that is neither the root nor at depth $k$, there exists a child $c$ of $v$ such that $T(c) \simeq T(p)|_d$, where $p$ is the parent of $v$ and $d$ is the depth of the subtree $T(c)$.*

*Proof.* We proceed by induction on $k$.

The base case $k = 1$ is clear: We have $\mathsf{cr}_1(T) = T$ for all $T \in \mathcal{T}_1$, so $\mathcal{C}_1 = \mathcal{T}_1$. On the other hand, the condition in Proposition A.2 becomes trivial since every node is the root or of depth $1$.

Let's do the induction step in both directions:

Let $T \in \mathcal{C}_{k+1}$, that is, $T = \mathsf{cr}_{k+1}(r_G)$ for some graph $G$ and vertex $r_G \in V(G)$. By construction, we have $T(c) \in \mathcal{C}_k$ for all $c \in N(r_T)$, where $r_T$ denotes the root of $T$. That is, by induction hypothesis all vertices at depth $k \geq 2$ have a child as desired. Let $v_T$ be a vertex at depth $1$ in $T$. By the definition of color refinement, there is a vertex $v_G \in N(r_G)$ such that $\mathsf{cr}_k(v_G) = T(v_T)$. Since $v_G$ and $r_G$ are adjacent, $v_T$ has a child $c$ such that $\mathsf{cr}_{k-1}(r_G) = T(c)$. That is, $T(c) = T|_{k-1}$ as desired.

In the other direction, suppose $T \in \mathcal{T}_{k+1}$ with all non-root vertices at depth less than $k + 1$ having a child $c$ satisfying $T(c) = T(p)|_d$. We show that there is a rooted tree $T'$ satisfying $\mathsf{cr}_{k+1}(T') = T$. Each child $v$ of the root $r_T$ has a child $c_v$ such that $T(c_v) = T|_{k-1}$. For each $v \in N(T)$ consider the subtree $T_v$ of $T(v)$ where $T(c_v)$ has been removed. By induction hypothesis there exists a rooted tree $T'_v$ such that $\mathsf{cr}_k(T'_v) = T_v$. Now construct the tree $T'$ by taking a root node $r$ with $x_r = x_T$ and connecting to it the tree $T_v$ via its root for each $v \in N(T)$. Then we have $\mathsf{cr}_{k+1}(T') = T$. $\qquad\square$

**Definition A.3.** For any random graph $G_t$ and $t \in \mathbb{N}$ we define the random PMF $c_{k,t}$ on $\mathcal{C}_k$ by

$$c_{k,t}(T) = t^{-1} \cdot |\{v \in V(G_t) : \mathsf{cr}_k(v) \simeq T\}|.$$

If $c_{k,t}$ converges in probability as $t \to \infty$, we denote its limit with $c_{k,\infty}$. If $c_{k,\infty}$ is defined, we call $G_t$ $\mathcal{C}_k$-*convergent*. If $G_t$ is $\mathcal{C}_k$-convergent for all $k \in \mathbb{N}$ we call $G_t$ *color convergent*.

## A.1 Color Convergence and Generalization Gap in MPNNs

**Example A.4.** $G_t$ is, with probability $1/2$, either a set of $t$ isolated vertices or a cycle on $t$ vertices. Consider the node label $f_*(v) = \mathbb{1}_{\{d_v = 0\}}$, which classifies isolated nodes. Let $f$ denote the constant $0$ classifier. Then $\mathbb{P}(R_{\mathrm{emp}}(f, G_t) = 0) = 1/2$ for $t \in \mathbb{N}$ but $R(f) = 1/2$.

**Theorem A.5.** *Let $G_t$ be a random graph. The following are equivalent:*

- *$G_t$ is $\mathcal{C}_k$-convergent.*

- *$G_t$ satisfies probabilistic consistency of empirical risk with respect to $k$-layer MPNNs.*

We divide the proof into Lemma A.6, which covers the forward implication, and Lemma A.7 and Lemma A.8, completing the equivalence.

**Lemma A.6.** *Suppose $G_t$ is $\mathcal{C}_k$-convergent. $G_t$ satisfies probabilistic consistency of empirical risk with respect to $k$-layer MPNNs.*

*Proof.* Since $G_t$ is $\mathcal{C}_k$-convergent, for every $T \in \mathcal{C}_k$ there exists $c_T \in [0, 1]$ such that for all $\varepsilon$ we have

$$\mathbb{P}(|c_{k,t}(T) - c_t| \leq \varepsilon) \to 1$$

as $t \to \infty$. Let

$$D := \{T \in \mathcal{C}_k : f(T) \neq f_*(\mathcal{C}_k)\}$$

denote the disagreement set. Then, for every $\varepsilon > 0$, we have

$$\mathbb{P}\left(\left|\sum_{T \in D}(c_{k,t}(T) - c_T)\right| \leq \varepsilon\right) \to 1$$

and thus

$$R_{\text{emp}}(f, G_t) = \sum_{T \in D} c_{k,t}(T) \qquad\qquad R(f) = \sum_{T \in D} c_T$$

and the empirical risk converges. $\qquad\qquad\qquad\qquad\qquad\qquad\qquad\qquad\square$

In the other direction we distinguish two ways in which $G_t$ can fail to be $\mathcal{C}_k$-convergent:

- Lemma A.7 captures the random graph failing to be sufficiently deterministic in the limit, as in Example A.4.
- The other mode of failure captures some probability mass escaping to infinity, covered by Lemma A.8. This happens for example in dense random graphs. "Tracking" the mass as it escapes to infinity with the empirical risk turns out to be more technically involved.

**Lemma A.7.** *Let $G_t$ be a random graph that. Suppose $c_{k,t}$ does not converge pointwise. That is, there exists $T \in \mathcal{C}_k$ such that for all $c \in [0,1]$ there exists $\varepsilon > 0$ such that*

$$\mathbb{P}(|c_{k,t}(T) - c| \leq \varepsilon) \not\to 1$$

*as $t \to \infty$. Then $G_t$ is does not satisfy probabilistic consistency of empirical risk with respect to $k$-layer GNNs.*

*Proof.* Let $f$ be constant $0$ and

$$f_*(v) = \begin{cases} 1 & \text{cr}_k(v) = T \\ 0 & \text{else} \end{cases}.$$

Then $R_{\text{emp}}(f, G_t) = c_{k,t}(T)$, which, by assumption, does not converge. $\qquad\qquad\square$

**Lemma A.8.** *Let $G_t$ be a random graph. Suppose $c_{k,t}$ does not converge in probability, but converges pointwise. That is, for every $T \in \mathcal{C}_k$ there exists $c_T$ such that for every $\varepsilon > 0$*

$$\mathbb{P}(|c_{k,t}(T) - c_T| \leq \varepsilon) \to 1$$

*as $t \to \infty$. Then $G_t$ is does not satisfy probabilistic consistency of empirical risk with respect to $k$-layer GNNs.*

*Proof.* Since $c_{k,t}$ converges point-wise to $c_T \in [0,1]$ but does not converge in probability, the mapping $T \mapsto c_T$ can not define a PMF. That is, there exists $\varepsilon_0 > 0$ such that

$$\sum_{T \in \mathcal{C}_k} c_T = 1 - \varepsilon_0.$$

Let $A_n \subseteq \mathcal{C}_k$ denote the set of refinement colors containing no node with more than $n$ children. Define

$$a_{n,t} := \sum_{T \in A_n} c_{k,t}(T).$$

and

$$a_{n,\infty} := \lim_{t \to \infty} a_{n,t} = \sum_{T \in A_n} c_T.$$

Note that for all $n \in \mathbb{N}, \varepsilon > 0$ we have

$$\mathbb{P}(|a_{n,t} - a_{n,\infty}| \leq \varepsilon) \to 1$$

as $t \to \infty$, $a_{n,\infty} \to 1 - \varepsilon_0$ as $n \to \infty$, and therefore

$$\mathbb{P}(|a_{n,t} - (1 - \varepsilon_0)|) \leq \varepsilon) \to 1$$

as $n \to \infty, t \to \infty$. Furthermore, for each $t \in \mathbb{N}$, choose $K_t \in \mathbb{N}$, such that

$$\mathbb{P}(a_{K_t,t} \geq 1 - 0.1\varepsilon_0) \geq 0.9.$$

The key idea now is to construct sets $A_{b_n}$ such that $A_{b_{n+1}} \setminus A_{b_n}$ "tracks" the mass $\varepsilon_0$ at time $b_{n+1}$ as it is escaping to infinity.

Let $N, t_0 \in \mathbb{N}$ such that for all $t \geq t_0$ we have $\mathbb{P}\left(|a_{N,t} - (1 - \varepsilon_0)| \leq 0.1\varepsilon_0\right) \geq 0.9$.
Construct the sequence $b_i$ over $\mathbb{N}$ as follows:

- Let $b_0 = K_{t_0}$.

- For $i \geq 1$, take $b_i > b_{i-1}$ such that

$$\mathbb{P}(a_{K_{b_i},b_i} - a_{K_{b_{i-1}},b_i} \geq 0.8\varepsilon_0) \geq 0.8.$$

Note that choosing such $b_i$ is always possible since

$$\mathbb{P}(a_{K_{b_{i-1}},t} \geq 1 - 0.9\varepsilon_0) \to 0$$

as $t \to \infty$.

Let $f$ be the constant $0$ function and define $f_*$ as follows:

$$f_*(v) = \begin{cases} 1 & \exists n \in \mathbb{N} : \mathsf{cr}_k(v) \in A_{b_{2n+1}} \setminus A_{b_{2n}} \\ 0 & \text{else} \end{cases}$$

The disagreement set $D$ is

$$D = \bigcup_{n \in \mathbb{N}} \left( A_{b_{2n+1}} \setminus A_{b_{2n}} \right).$$

Then we have

$$\mathbb{P}(|R_{\mathrm{emp}}(f, G_{b_{2n+1}})|) \geq 0.8\varepsilon_0) \geq \mathbb{P}(a_{K_{b_{2n+1}},b_{2n+1}} - a_{K_{b_{2n}},b_{2n+1}} \geq 0.8\varepsilon_0) \geq 0.8$$

$$\begin{aligned}
\mathbb{P}(|R_{\mathrm{emp}}(f, G_{b_{2n+2}})|) \leq 0.3\varepsilon_0) &\geq \mathbb{P}(a_{N,b_{2n+2}} + (a_{K_{b_{2n+2}},b_{2n+2}} - a_{K_{b_{2n+1}},b_{2n+2}}) \geq 1 - 0.3\varepsilon_0) \\
&\geq \mathbb{P}(a_{N,b_{2n+2}} \geq 1 - 1.1\varepsilon_0 \wedge a_{K_{b_{2n+2}},b_{2n+2}} - a_{K_{b_{2n+1}},b_{2n+2}} \geq 0.8\varepsilon_0) \\
&\geq \mathbb{P}(a_{N,b_{2n+2}} \geq 1 - 1.1\varepsilon_0) + \mathbb{P}(a_{K_{b_{2n+2}},b_{2n+2}} - a_{K_{b_{2n+1}},b_{2n+2}} \geq 0.8\varepsilon_0) - 1 \\
&\geq 0.7
\end{aligned}$$

for all $n \in \mathbb{N}$. That is, the empirical risk does not converge. $\qquad\square$

Corollary A.9 follows immediately from the definitions.

**Corollary A.9.** *Let $G_t$ be a random graph. Then the following are equivalent:*

- *$G_t$ is color convergent.*

- *$G_t$ satisfies probabilistic consistency of empirical risk with respect to MPNNs.*

## A.2 Properties of Color Convergent Random Graphs

**Proposition A.10.** *Let $G_t$ be a $\mathcal{B}_k$-convergent random graph. Then $G_t$ is $\mathcal{C}_k$-convergent.*

*Proof.* For $T \in \mathcal{C}_k$ define $A_T := \{B \in \mathcal{B}_k : \mathsf{cr}_k(B) = T\}$ and

$$c_T = \sum_{B \in A_T} b_{k,\infty}(B).$$

Then $T \mapsto c_T$ defines a PMF and, for every $\varepsilon > 0$, we have

$$\mathbb{P}(|c_{k,t}(T) - c_T| \leq \varepsilon) = \mathbb{P}\left( \left| \sum_{B \in A_T} (b_{k,t}(B) - b_{k,\infty}(B)) \right| \leq \varepsilon \right) \to 1$$

as $t \to \infty$. $\qquad\square$

**Proposition A.11.** *Suppose $G_t$ is a locally tree-like and $\mathcal{C}_k$-convergent. Then $G_t$ is $\mathcal{B}_k$-convergent and, for $T \in \mathcal{T}_k$, we have*

$$b_{k,\infty}(T) = c_{k,\infty}(\mathsf{cr}_k(T)).$$

*Proof.* Let $\varepsilon, \delta \geq 0$. Since $G_t$ is locally tree-like, there exists $N_{\text{tree}}$ such that for $t \geq N_{\text{tree}}$ we have

$$\mathbb{P}\left(t^{-1} \cdot |\{v \in G_t : B_k(v) \text{ contains a cycle}\}| \leq \varepsilon/2\right) \geq 1 - \delta/2.$$

Let $T \in \mathcal{C}_k$. Since $G_t$ is $\mathcal{C}_k$ convergent, there exists $N$ such that for $t \geq N$ we have

$$\mathbb{P}\left(|c_{k,t}(\mathsf{cr}_k(T)) - c_{k,\infty}(\mathsf{cr}_k(T))| \leq \varepsilon/2\right) \geq 1 - \delta/2.$$

Let $t \geq \max(N_{\text{tree}}, N)$. Then we have

$$\mathbb{P}(|t^{-1} \cdot |\{v \in G_t : \mathsf{cr}_k(v) = \mathsf{cr}_k(T) \wedge B_k(v) \text{ is a tree}\}| - c_{k,\infty}(\mathsf{cr}_k(T))| \leq \varepsilon) \geq 1 - \delta.$$

Since $\mathsf{cr}_k(v) = \mathsf{cr}_k(T)$ together with $B_k(v)$ being a tree implies $B_k(v) = T$ we are done. $\square$

**Definition A.12.** A PMF $\mu$ on $\mathcal{C}_k$ is *sofic* if there exists a random graph $G_t$ such that $\mu = c_{k,\infty}$.

**Definition A.13.** Let $\mu$ be a PMF on $\mathcal{C}_k$ and define

$$d_\mu := \sum_{T \in \mathcal{C}_k} d_T \cdot \mu(T).$$

If $d_\mu < \infty$, we define the *edge-type marginal* $\bar{\mu}$, a PMF on $\mathcal{C}_{k-1}^2$, as

$$\bar{\mu}(T_0, T_1) := \frac{1}{d_\mu} \cdot \sum_{\substack{T \in \mathcal{C}_k \\ T|_{k-1} = T_0}} |\{c \in N(T) : T(c) = T_1\}| \cdot \mu(T).$$

If $d_\mu < \infty$ and $\bar{\mu}$ is symmetric we call $\mu$ *involution invariant with finite degree*.

**Theorem A.14.** *Let $k \geq 2$, $\mu$ a sofic PMF on $\mathcal{C}_k$. Then $\mu$ is involution invariant with finite degree.*

*Proof.* Consider a random graph $G_t$ such that $c_{k,\infty} = \mu$.

Define the *excess* $c'_{k,t}$ as the PMF on $\mathcal{C}_{k-1}$ of the following sampling process:

- Sample $T \sim c_{k,t}$.
- Uniformly at random chose a node $c$ among the children of the root.
- Return $T(c)$.

By construction, $c'_{k,t}(T)$ is proportional to $d_T \cdot c_{k-1,t}(T)$, More concretely, we have

$$d_{c_{k,t}} \cdot c'_{k,t}(T) = d_T \cdot c_{k-1,t}(T)$$

for all $T \in \mathcal{C}_{k-1}$. Furthermore, $c'_{k,\infty}$ is a well-defined PMF and we have $c'_{k,t}(T) \to c'_{k,\infty}(T)$ as $t \to \infty$. It follows that

$$d_{c_{k,\infty}} \cdot c'_{k,\infty}(T) = d_T \cdot c_{k-1,\infty}(T)$$

for all $T \in \mathcal{C}_{k-1}$. That is, $d_\mu = d_{c_{k,\infty}} < \infty$.

The symmetry of $\bar{\mu}$ then follows from the symmetry of $\bar{c}_{k,t}$, the convergence of $c_{k,t}$ to $\mu$, and the bound

$$|\{c \in N(T) : T(c) = T_1\}| \leq d_{T_0}$$

for $T \in \mathcal{C}_k$ with $T|_{k-1} = T_0$. $\square$

**Corollary A.15.** *Let $k \geq 2$ and suppose $G_t$ is $\mathcal{C}_k$-convergent. Then $\mathbb{E}[|E(G_t)|] \in O(t)$.*

*Proof.* We have $\mathbb{E}[|E(G_t)|] = t \cdot \frac{d_{c_{k,t}}}{2}$ and $d_{c_{k,t}} \in O(1)$. $\square$

# B   Refined Configuration Model Proofs

**Definition B.1.** The *refined configuration model* $\mathsf{RCM}_t(\mu)$ is parametrized by:

- a finite or countable set of types $S$, with a type-to-feature mapping $s \to x_s$,

- a PMF $\mu$ over $S \times \mathrm{Multiset}(S)$, the product of types and finite multisets of types.

$\mathsf{RCM}_t(\mu)$ is defined on $\{v_i\}_{i \in [t]}$ as follows:

- For each node $v_i$ assign a type-multiset pair $(s_i, A_i) \sim \mu$ independently at random. $s_i$ determines the type of $v_i$, while $A_i$ determines the types of nodes $v_i$ may be connected to.

- For each type $s \in S$, we independently generate a configuration model on the vertices of type $s$:

    - Each vertex $v_i$ with $s_i = s$ is given a stub for each occurrence of $s$ in $A_i$. The stubs are paired uniformly at random to form edges, until there are 0 or 1 stubs left.

- For distinct types $s_L \neq s_R$, we independently generate a bipartite configuration model on the vertices of type $s_L$ and $s_R$:

    - Each vertex $v_i$ with $s_i = s_L$ is assigned to the set $L$ of left nodes. Each vertex $v_i$ with $s_i = s_R$ is assigned to the set $R$ if right nodes.
    - Each $v_i \in L$ is given a stub for each occurrence of $s_R$ in $A_i$. Each $v_i \in R$ is given a stub for each occurrence of $s_L$ in $A_i$. Then the left stubs are matched uniformly at random with the right stubs to form edges, until there are no more stubs left in $L$ or $R$.

Our model is essentially not more complex than a combination of configuration and bipartite configuration models, which are well-behaved under the conditions given in Definition B.2.

**Definition B.2.** Let $\mu$ be a PMF on $S \times \mathrm{MultiSet}(S)$ and define

$$d_\mu := \sum_{s \in S} \sum_{A \in \mathrm{MultiSet}(S)} |A| \cdot \mu(s, A).$$

If $d_\mu < \infty$, we define the *edge-type marginal* $\bar{\mu}$, a PMF on $S^2$, as

$$\bar{\mu}(s_0, s_1) := \frac{1}{d_\mu} \cdot \sum_{A \in \mathrm{MultiSet}(S)} |\{\!\{a \in A : a = s_1\}\!\}| \cdot \mu(s_0, A).$$

If $d_\mu < \infty$ and $\bar{\mu}$ is symmetric we call $\mu$ *involution invariant with finite degree*.

Definition B.2 ensures that, for involution invariant $\mu$ with finite degree, the subgraph of $\mathsf{RCM}_t(\mu)$ spanned by the nodes $U_s$ of a given type $s \in S$ is equal to $\mathsf{CM}(\nu)$, where

$$\nu(n) = \frac{1}{Z_s} \sum_{\substack{A \in \mathrm{MultiSet}(S) \\ \{\!\{s' \in A : s' = s\}\!\} = n}} \mu(s, A) \qquad\qquad Z_s := \sum_{A \in \mathrm{MultiSet}(S)} \mu(s, A)$$

and the bipartite subgraph of $\mathsf{RCM}_t(\mu)$ between nodes $U_{s_L}$ and $U_{s_R}$ for distinct types $s_L, s_R \in S$ is equal to $\mathsf{BCM}(\nu_L, \nu_R)$ with

$$\nu_L(n) = \frac{1}{Z_{s_L}} \sum_{\substack{A \in \mathrm{MultiSet}(S) \\ \{\!\{s' \in A : s' = s_R\}\!\} = n}} \mu(s_L, A) \qquad\qquad \nu_R(n) = \frac{1}{Z_{s_R}} \sum_{\substack{A \in \mathrm{MultiSet}(S) \\ \{\!\{s' \in A : s' = s_L\}\!\} = n}} \mu(s_R, A)$$

Given the corresponding results for these models given e.g. in Van Der Hofstad (2024), and due to the independence of the edge sampling procedures, it follows that $\mathsf{RCM}_t(\mu)$ converges locally to the following Galton-Watson tree:

**Theorem B.3.** *Suppose $G_t = \mathsf{RCM}_t(\mu)$ parametrized by types $S_0$, type-to-feature mapping $s \to x_s$ and is involution invariant $\mu$ with finite degree. Then $G_t$ converges locally to the GWT $W_t$ parametrized by type set $S = (\{\perp\} \cup S_0) \times S_0$, type-to-feature mapping $(s_0, s_1) \mapsto x_{s_1}$ and*

$$\mu_0(\perp, s) = Z_s \qquad\qquad \mu_0(s_0, s_1) = 0 \tag{6}$$

$$\mu_{\perp,s}(A) = \begin{cases} \frac{1}{Z_s}\mu(s, \{\!\{q : (p, q) \in A\}\!\}) & \forall (p, q) \in A : p = s \\ 0 & otherwise \end{cases} \tag{7}$$

$$\mu_{s_0,s_1}(A) = \frac{|\{\!\{s \in A : s = (s_1, s_0)\}\!\}| \cdot \mu_{\perp,s_1}(A)}{\displaystyle\sum_{B \in MultiSet(S)} |\{\!\{s \in B : s = (s_1, s_0)\}\!\}| \cdot \mu_{\perp,s_1}(B)}. \tag{8}$$

*for all $s_0, s_1 \in S_0$.*

Intuitively, the state-pair $(s_0, s_1)$ of each node is aware of its own state $s_1 \in S_0$, but also its parent's state $s_0 \in S_0$. Root nodes have no parent (6). Given a root $r$, $\mu_{\perp,r}(A)$ is given by $\mu(s, A)$ conditioned on $s = r$, and each element $(p, q)$ of $A$ has to come with the correct parent node $p = r$ (7). Finally, $\mu_{s_0,s_1}$ is simply $\mu_{\perp,s_1}$ conditioned on there being a neighbor of type $s_0$ (8). For a more detailed intuition refer to Definition B.7. In fact, we shall see in Theorem B.11 that every GWT that arises as the local limit of a random graph can be represented in this way.

Corollary B.4 is a direct consequences of Theorem B.3.

**Corollary B.4.** *Let $G_t = \mathsf{RCM}_t(\mu)$, $S = X$ and $x_s = s$. If $\mu$ is involution invariant with finite degree,*

- *$G_t$ is locally tree-like.*

- *$c_{1,\infty}(T) = \mu(x_T, \{\!\{x_v : v \in N(T)\}\!\})$ for all $T \in \mathcal{C}_1$.*

*Proof.* $G_t$ being tree-like is clear. Furthermore, we have

$$c_{1,\infty}(T) = \mu_0(\perp, x_T) \cdot \mu_{\perp,x_T}(\{\!\{(x_T, x_v) : v \in N(T)\}\!\}) = \mu(x_T, \{\!\{x_v : v \in N(T)\}\!\})$$

for $T \in \mathcal{C}_k$. $\qquad\square$

**Corollary B.5.** *Suppose $\nu$ is a sofic distribution over $\mathcal{C}_k$. Then there is a refined configuration model $G_t = \mathsf{RCM}_t(\mu)$ with types $S = \mathcal{C}_{k-1}$ such that $c_{k,\infty} = \nu$.*

*Proof.* Let

$$\mu(s, A) = \begin{cases} \nu(T) & \text{if there exists } T \in \mathcal{C}_k \text{ such that } T|_{k-1} = s \text{ and } \{\!\{T(c) \mid c \in N(T)\}\!\} = A \\ 0 & \text{otherwise} \end{cases}.$$

By Corollary B.4, for every $\varepsilon > 0$, we have

$$\mathbb{P}\left(t^{-1} \cdot |\{v_i \in V(G_t) : \{\!\{s_{v_j} : v_j \in N(v_i)\}\!\} = A_i\}| \geq 1 - \varepsilon\right) \to 1$$

as $t \to \infty$. Setting the type-to-feature mapping $s \to x_s$, we obtain

$$\mathbb{P}\left(t^{-1} \cdot |\{v_i \in V(G_t) : \{\!\{\mathsf{cr}_{k-1}(v_j) : v_j \in N(v_i)\}\!\} = A_i\}| \geq 1 - \varepsilon\right) \to 1$$

as $t \to \infty$. It follows that $c_{k,\infty} = \nu$. $\qquad\square$

**Corollary B.6.** *Let $G_t$ be a random graph. The following are equivalent:*

- *$G_t$ satisfies probabilistic consistency of empirical risk with respect to $k$-layer MPNNs*

- *There exists an involution invariant PMF $\mu$ with finite degree such that $\mathsf{RCM}_t(\mu)$ is equivalent to $G_t$ in probability for all $k$-layer MPNNs $f$. That is, for all $\varepsilon > 0$, as $t \to \infty$,*

$$P(|R_{emp}(f, \mathsf{RCM}_t(\mu)) - R_{emp}(f, G_t)| \geq \varepsilon) \to 0.$$

*Proof.* Suppose $G_t$ satisfies probabilistic consistency of empirical risk with respect to $k$-layer MPNNs. By Theorem A.5 there exists a PMF $\nu$ over $\mathcal{C}_k$ such that $c_{k,\infty} = \nu$. By A.14 this PMF is involution invariant with finite degree. By Corollary B.5 there exists an involution invariant distribution $\mu$ with finite degree such that setting $G'_t = \mathsf{RCM}_t(\mu)$ yields $c_{k,\infty} = c'_{k,\infty}$. Let $f, f_*$ be any $k$-layer GNN and define

$$D := \{T \in \mathcal{C}_k : f(T) \neq f_*(T)\}$$

the disagreement set. Since $c_{k,\infty} = c'_{k,\infty}$ we have

$$\mathbb{P}(|R_{\mathrm{emp}}(f, \mathsf{RCM}_t(\mu)) - R_{\mathrm{emp}}(f, G_t)| \geq \varepsilon) = \mathbb{P}\left(\left|\sum_{T \in D}(c'_{k,t}(T) - c_{k,t}(T))\right| \geq \varepsilon\right) \to 0$$

as $t \to \infty$.

Suppose on the other hand there exists an involution invariant $\mu$ with finite degree and $G'_t = \mathsf{RCM}_t(\mu)$ such that for all $\varepsilon > 0$ and $k$-layer MPNNs $f$,

$$\mathbb{P}(|R_{\mathrm{emp}}(f, \mathsf{RCM}_t(\mu)) - R_{\mathrm{emp}}(f, G_t)| \geq \varepsilon) = \mathbb{P}\left(\left|\sum_{T \in D}(c'_{k,t}(T) - c_{k,t}(T))\right| \geq \varepsilon\right) \to 0$$

as $t \to \infty$. Then, in particular, this is true for $f$ constant $0$ and $f_*$ the indicator function of $T$. That is, for all $T \in \mathcal{C}_k$ and $\varepsilon > 0$, we have

$$\mathbb{P}\left(\left|c'_{k,t}(T) - c_{k,t}(T)\right| \geq \varepsilon\right) \to 0$$

as $t \to \infty$. Since $G'_t = \mathsf{RCM}_t(\mu)$ is color convergent, $c_{k,t}$ converges in probability to $c'_{k,\infty}$. $\square$

For the final result we need to show that every GWT can be represented as in Theorem B.3.

**Definition B.7.** A *simplified unimodular* GWT $W_t$ is a GWT parametrized by $S, \mu_0, \{\mu_s\}_{s \in S}$ such that

- there is a base type set $S_0$ such that $S = (\{\perp\} \cup S_0) \times S_0$, that is, each node's type records its own base type and its parent's base type ($\perp$ in the case of the root).

- The root distribution $\mu_0$ is supported on nodes of type $(\perp, s)$, i.e., $\mu_0(s_0, s_1) = 0$ if $s_0 \neq \perp$.

- For all $s_0, s_1 \in S_0$, the offspring distribution $\mu_{s_0, s_1}$ is supported only on multisets of children of the form $\{\{(s_1, q_i)\}_{i=1}^n\}$, meaning every child has parent type $s_1$:

$$\mu_{s_0, s_1}(A) = 0 \quad \text{if } \exists\, (p, q) \in A \text{ with } p \neq s_1.$$

Additionally, the following two conditions hold:

- (Root-to-child consistency) For all $s_0 \neq \perp$, the pmf $\mu_{s_0, s_1}$ coincides with the conditional offspring distribution of a root with type $(\perp, s_1)$ and conditioned on having a child of type $(s_1, s_0)$, that is

$$\mu_{s_0, s_1}(A) = \frac{|\{\!\{s \in A : s = (s_1, s_0)\}\!\}| \cdot \mu_{\perp, s_1}(A)}{\sum\limits_{B \in \mathrm{MultiSet}(S)} |\{\!\{s \in B : s = (s_1, s_0)\}\!\}| \cdot \mu_{\perp, s_1}(B)}.$$

- (Edge-type symmetry) Let the expected degree of the tree be

$$d_W := \sum_{s \in S_0} \sum_A |A| \cdot \mu_0(\perp, s) \cdot \mu_{\perp, s}(A),$$

and define the edge-type marginal

$$\bar{\mu}(s_0, s_1) := \frac{1}{d_W} \sum_A |\{a \in A : a = s_1\}| \cdot \mu_0(\perp, s_0) \cdot \mu_{\perp, s_0}(\{\!\{s_0, a\}\!\}_{a \in A}).$$

Then we require that $\bar{\mu}$ is symmetric:

$$\bar{\mu}(s_0, s_1) = \bar{\mu}(s_1, s_0).$$

Note that $S_0$, $\mu_0$, $\{\mu_{(\perp,s)}\}_{s \in S_0}$ completely determine a simplified unimodular GWT. The GWT in Theorem B.3 is an simplified unimodular GWT as edge-type symmetry is a direct consequence of involution invariance with finite degree.

**Lemma B.8.** *Suppose $\mu$ is involution invariant with finite degree. Then $\mathsf{RCM}_t(\mu)$ converges locally to a simplified unimodular Galton-Watson tree.*

In fact, Definition B.7 exactly captures the local limits of refined configuration models.

**Lemma B.9.** *Suppose $W_t$ is a simplified unimodular GWT determined by $S_0$, $\mu_0$, $\{\mu_{(\perp,s)}\}_{s \in S_0}$. Then $W_t$ is the local limit of $\mathsf{RCM}_t(\mu)$ where $\mu$ is involution invariant with finite degree.*

*Proof.* Define the distribution $\mu$ over $S_0 \times \mathsf{MultiSet}(S)$ by

$$\mu(s, A) = \mu_0(\perp, s) \cdot \mu_{\perp,s}(\{\!\{(s,a) : a \in A\}\!\}).$$

Involution invariance with finite degree of $\mu$ follows directly from edge-type symmetry of $W_t$. Applying Theorem B.3 we recover $W_t$ as the local limit of $\mathsf{RCM}_t(\mu)$. $\qquad\square$

All that's left is to show that every sofic GWT is equivalent to a simplified, unimodular GWT.

**Lemma B.10.** *Suppose $W_t'$ is a Galton Watson tree that is the local limit of a random graph. There exists a simplified unimodular Galton Watson tree $W_t$ that defines the same random process.*

*Proof.* Suppose $W_t'$ is parametrized by $S'$, $\mu_0'$. $\{\mu_s'\}_{s \in S'}$ and type-to-feature mapping $s \to x_s'$, and is the local limit of $G_t$. Define

$$S_0 := \{s \in S' : \mu_0'(s) > 0\},$$

the set of vertices which appear with positive probability at the root.

Due to unimodularity Aldous & Lyons (2007), the nodes that appear as the neighbors of the root with non-zero probability, must themselves appear as the root with non-zero probability as well. Therefore this set of types is sufficient. Let us make this more formal:

Let $\mu_k^s$ denote the PMF of the conditional distribution of $W_k'$ given that the root has state $s$. Fix an total order $\preceq$ on $S_0$. For every $s \in S_0$, there is a distribution $\nu^s$ over $\mathsf{MultiSet}(S_0)$ such that, for every $k \in \mathbb{N}$, sampling $T \sim \mu_k^s$ is equivalent to the following process:

- Consider a singleton graph comprising the root $r$ with feature $x_r = x_s$.

- Sample a multiset $\{\!\{s_0 \preceq \cdots \preceq s_n\}\!\} \sim \nu^s$.

- Sample $B_0, \ldots, B_n \sim \mu_{k-1}^{s_0} \times \cdots \times \mu_{k-1}^{s_n}$, conditioned on the existence of a neighbor $v_i$ of $B_i$ such that $x_{v_i} = x_r$ and all the balls agreeing on the neighborhood of $v_i$, that is, $B_{k-2}(v_i) = B_{k-2}(v_j)$ for $i, j \in [n+1]$.

- Connect the graphs $B_i$ to the root $r$ via the neighbor whose existence we conditioned on, and return the resulting tree.

Setting $\mu_0(\perp, s) = \mu_0(s)$ and

$$\mu_{\perp,s}(A) = \begin{cases} \nu^s(\{\!\{q : (p,q) \in A\}\!\}) & \forall (p,q) \in A : p = s \\ 0 & \text{otherwise} \end{cases}$$

for $s \in S_0$ we obtain the desired simplified unimodular GWT. $\qquad\square$

Theorem B.11 now follows directly from Lemma B.8, Lemma B.9 and Lemma B.10.

**Theorem B.11.** *The following are equivalent:*

- *$W_t$ is the local limit of $\mathsf{RCM}_t(\mu)$ for some involution invariant PMF $\mu$ with finite degree.*

- *$W_t$ is a GWT that arises as the local limit of some random graph.*

