# OpenReview forum: "On Local Limits of Sparse Random Graphs: Color Convergence and the Refined Configuration Model"
_NeurIPS.cc/2025/Conference — NeurIPS 2025 poster_

### Official Review · Reviewer_wyHK · 2025-07-01

**Clarity:** 3
**Significance:** 3
**Originality:** 3
**Rating:** 4
**Confidence:** 1

**Summary:**

The paper proposes color convergence, a new graph-limit notion aligned with 1-WL color refinement, and shows it is equivalent to uniform generalization of every $k$-layer MPNN on node-classification tasks. To generate all such limits, the authors introduce the Refined Configuration Model, a type-augmented configuration model that subsumes E-R, SBM, and classical configuration graphs while remaining locally tree-like.

**Questions:**

Apart from the comments in the weakness section, I have the following questions:
1. The framework focuses on node classification. Could the framework be extended to provide insights into other graph learning tasks?
2. How do the authors expect one would fit the parameters (e.g., the PMF $\mu$) from an observed real-world sparse graph? Are there consistent estimators, perhaps leveraging methods of moments on refinement-color counts, that could be used in practice?
3. Why only 2 colors in this work? Could the authors extend their framework to multiple colors?

**Ethical Concerns:**

["NO or VERY MINOR ethics concerns only"]

**Final Justification:**

The authors' response cleared my doubts, and I now understand the potential applications of this work. Therefore, I raised my score to 4.

**Limitations:**

Yes.

**Quality:**

3

**Strengths And Weaknesses:**

Strengths:
1. First limit concept tied to MPNNs with bridge between graph limits and GNN learning theory.
2. Proofs of equivalence, universality, and involution-invariance look reasonable, although I did not check the proofs carefully.

Weaknesses:
1. Applications of this work are unclear to me.
2. The contribution is purely theoretical; no empirical or synthetic experiments are provided.

---

> ### Author Rebuttal · Authors · 2025-07-30
>
> We thank the reviewer for their feedback and interesting questions.
>
> **[W1&2]** A major contribution of **our work is furthering the understanding of (learnable) graph data models in the context of MPNNs**. This required taking an abstract, non-parametric approach. Concrete experiments would ultimately require arbitrary parametrization choices, which may not be representative of our more general results. We will highlight the relevance of our work to the machine learning community (see reply to reviewer 1zSd, W2) and point out existing empirical evidence for our results (see reply to reviewer X2fm, W3).
>
> **[Q1]** Many approaches to tasks such as graph classification and link prediction utilize message-passing layers within their architectures, and **our theoretical results apply to any such message-passing components**. Extending our results to end-to-end analysis of such composite architectures is an interesting direction for future work.
>
> **[Q2]** The RCM exhibits very diverse behavior depending on the parametrization (e.g. sparse SBM if $\mu$ is multivariate Poisson, bipartite configuration mode if there are 2 types and $\mu(s, A) = 0$ for $s\in A$, etc.). In our view, the choice of parametrization should be informed by the application domain. Molecular structures may benefit from discrete distributions reflecting their rigid topology. Citation networks and other growing graphs suggest counting distributions, though whether Poisson or alternatives like power-law distributions are optimal remains context-dependent. Some applications may require infinite type spaces or mixed approaches. Investigating specific parametrizations and their properties, e.g. existence of consistent estimators, represents important future work.
>
> **[Q3]** The two-color examples in the paper were chosen only for illustrative purposes. **All results are stated for finite or countably infinite color spaces**.

---

### Official Review · Reviewer_HKEY · 2025-07-03

**Clarity:** 3
**Significance:** 3
**Originality:** 4
**Rating:** 5
**Confidence:** 3

**Summary:**

This paper studies the local limits of sparse random graphs, proposes a notion of local convergence, and also a random graph model. The proposed color convergence generalizes the local convergence and draws a strong relation to the learnability of Message Passing Graph Neural Networks (MPNN). Color convergence is defined based on the classic color refinement algorithm by Weisfeiler and Leman. The proposed Refined Configuration Model (RCM) generalizes the configuration model and it is universal with respect to coloar convergence and local convergence to Galton-Watson trees.

**Questions:**

I am curious about is there any arguments about weak local limits we can make using the proposed framework? By weak local limit, I am referring to the Definition 2.1 in [1] for example.


[1] Berger, Noam, et al. "Asymptotic behavior and distributional limits of preferential attachment graphs." The Annals of Probability (2014): 1-40.

**Ethical Concerns:**

["NO or VERY MINOR ethics concerns only"]

**Final Justification:**

My opinion on this paper is very positive and the rebuttal of the authors to my question is satisfactory. I keep my score as it is.

**Limitations:**

Yes.

**Paper Formatting Concerns:**

No concerns.

**Quality:**

3

**Strengths And Weaknesses:**

1. This paper is well-organized and easy to follow. Key concepts are defined and delivered clearly. There are also quite a few figures that enhance the understandability.
2. The proposed color convergence notion exactly capture the learnability of MPNN in the sense that there is a necessary and sufficient condition on whether a random graph satisfies probabilistic consistency of empirical risk with respect to $k$-layer MPNNs. Although this is not a very strong consistency guarantee, it opens up future directions on deriving stronger guarantees such as specific error bounds.
3. RCM generalizes the commonly studied configuration model. It has potential to be studied in many follow-up works that concern with MPNNs or need a more general random graph with a nice local limit structure.

---

> ### Author Rebuttal · Authors · 2025-07-30
>
> We appreciate the reviewer for finding our work well-organized and easy to follow, and thank them for highlighting its key features in the review.
>
> **[Q1]** Note that color refinement $\mathsf{cr}\_k$ defines a one-to-one mapping between the set $\mathcal{T}\_k$ of rooted trees of depth $k$, and the set $\mathcal{C}\_k$ of refinement colors of depth $k$. Hence, for each color convergent random graph $G\_t$ its limit distributions of refinement colors $c_{k, \infty}$ give rise to a unique random rooted tree $W_k$, such that the probability $c_{k, t}(T)$ converges to $P(\mathsf{cr}\_k(W_k) = T)$ as $t\to\infty$ for every refinement color $T\in\mathcal{C}_k$. This is the strongest possible characterization: Since non-isomorphic neighborhoods can share the same refinement color, **a local weak limit in terms of isomorphic neighborhoods may not exist**.
>
> While the example in Remark 3.9 fails to converge locally according to our definition (i.e. convergence in probability), a local weak limit (i.e. convergence in distribution) still exists. We thank the reviewer for directing our attention to this. **We will strengthen Remark 3.9 to also address local weak convergence** by redefining the random graph $G_t$ as follows:
>
> + If $t$ is a multiple of $3$, $G_t$ deterministically consists of $t/3$ different $3$-cycles.
> + Otherwise, $G_t$ deterministically consists of a single $t$-cycle.
>
> As before, $G_t$ is deterministically 2-regular and therefore color convergent. However, a local weak limit no longer exists.

---

> > ### Comment · Reviewer_HKEY · 2025-08-06
> >
> > I appreciate the clarification.

---

### Official Review · Reviewer_X2fm · 2025-07-03

**Clarity:** 4
**Significance:** 3
**Originality:** 3
**Rating:** 5
**Confidence:** 3

**Summary:**

Local convergence characterizes the neighbourhood distribution around a randomly chosen node in the large‑graph limit, providing several structural details. This is exploited by the Weisfeiler–Leman (WL) color‑refinement algorithm that bounds the expressive power of Message Passing Graph Neural Networks (MPNNs). Using this observation, they propose color convergence, a refinement that facilitates the analysis of MPNNs through the framework of local convergence.

They prove that a random graph sequence is color-convergent if and only if it satisfies probabilistic consistency of empirical risk with respect to k-layer MPNNs. They then designed the Refined Configuration Model (RCM) that is universal with respect to color convergence. It is a type-labelled extension of the classical configuration model. This framework helps them develop structural results for locally tree-like graphs. Through this framework they show that colour-convergent sequences correspond precisely to local limits given by Galton–Watson trees.

**Questions:**

Typo: not is written twice in Remark 2.9

A line before Corollary 3.14 explaining it would be nice. Took me a while to see.

**Ethical Concerns:**

["NO or VERY MINOR ethics concerns only"]

**Final Justification:**

I am updating my score as the issues I raised were addressed in the rebuttal.

**Limitations:**

Yes

**Paper Formatting Concerns:**

None.

**Quality:**

3

**Strengths And Weaknesses:**

Strengths
- Bridging WL-based expressivity analysis of GNNs with graph-limit theory is new and interesting. The idea of using Color convergence for this is a good balance. It is weaker than local convergence but captures learnability by MPNNs.
- The writing is clean. Related work is covered well and defintions are clearly stated with examples.


Weakness:
- They do not give intutions for their proofs.
- No formal guarantees regarding convergence rates or error bounds.
- Given that colour convergence is a new framework they are proposing aimed at explaining MPNN behaviour, even a small controlled simulation would greatly strengthen the paper.

---

> ### Author Rebuttal · Authors · 2025-07-30
>
> We thank the reviewer for their insightful comments and for appreciating our work.
>
> **[W1]** We acknowledge that space constraints limited our explanations. We plan to use the extra page in the final version to provide additional intuition.
>
> **[W2]** Please see reply to reviewer 1zSd, W3. **Color-convergence and RCMs are  very general and broad frameworks which can potentially encode very diverse convergence phenomena**. Analyzing convergence rates and error bounds for different parameterizations and in general will form interesting future work.
>
> **[W3]** **Empirical evidence for the impossibility of learning on random graphs that are not color convergent exists in literature for restricted cases [2, 4]**. We will highlight existing empirical evidence for Theorem 3.6 (also see reply to Reviewer 1zSd).
> Generality of our results critically relies on taking a non-parametric approach. Any simulation would ultimately require arbitrary parametric choices, which may not be representative of our more general results.
>
> **[Q1]** We thank the reviewer for noticing the typo and will fix it in the final manuscript.
>
> **[Q2]** We thank the reviewer for directing our attention to this. We will add a sentence before Corollary 3.14 connecting the finiteness of $d_\mu$ to the boundedness of average degree, and thus to the linear growth of edge count.
>
> [2] Yehudai, G., Fetaya, E., Meirom, E., Chechik, G., & Maron, H. (2021). From local structures to size generalization in graph neural networks. *38th International Conference on Machine Learning.*
>
> [4] Adam-Day, S., Benedikt, M., Ceylan, I., & Finkelshtein, B. (2024). Almost surely asymptotically constant graph neural networks. *Advances in Neural Information Processing Systems, 37.*

---

### Official Review · Reviewer_1zSd · 2025-07-08

**Clarity:** 2
**Significance:** 3
**Originality:** 3
**Rating:** 3
**Confidence:** 3

**Summary:**

The paper introduces a notion of "local convergence" that they call "color convergence" and also proposes what the authors call the "refined configuration model", which is a variant of the configuration model used in random graphs. This can be motivated as follows to an ML audience: one is often interested in understanding if a local aggregation algorithm, such as a graph neural network based on message passing, ends up "performing well" on a graph drawn from a probabilistic model. Here, one can ask if there is a general characterization under which the "recovery" up to an \epsilon error in the loss objective occurs with high probability. For procedures that fall under the umbrella of the Weisfeiler-Leman algorithm (including GNNs), the authors show that "color convergence" provides such a characterization.

Color convergence is a local-convergence style property that is based on the distribution of trees obtained via a color refinement algorithm. The main result is then that if a random graph family is color convergent, then with high probability, an MPNN procedure on a graph drawn from the family achieves a small generalization gap (difference between the true risk and the empirical risk).

**Questions:**

Please comment on each of the points in the weaknesses section. Also, could you discuss more on the significance of some of the results like 3.13 and the definitions around it?

**Ethical Concerns:**

["NO or VERY MINOR ethics concerns only"]

**Final Justification:**

I have read the authors' rebuttal, and while they acknowledge and address the concerns to a certain extent, there are still gaps that require significant re-writing imo. I prefer to keep my score.

**Limitations:**

yes

**Quality:**

3

**Strengths And Weaknesses:**

Strengths:

- The problem of achieving small generalization gap via GNN type procedures on classes of random graphs is an important one, as it gives insights into when one can expect these methods to work well on real graphs.
- The results obtained are quite interesting, assuming they can be motivated appropriately to an ML audience.
- The variant of the configuration model seems interesting as well.

Weaknesses:

- The biggest concerns about the paper are about the writing and how it does not make the case for "fit" for an ML conference like NeurIPS. Right now, the paper reads like one that is of interest to the graph limits community, but perhaps not much beyond.

- Even though the paper gives a few concrete examples of random graph models, color refinement, etc., the writing is far from clear. Suppose we focus on applications of Theorem 3.6, are there specific graph classes on which GNN approaches were not known to be consistent but the new result sheds light? (I imagine there are, but it would be good to see examples.)

- The authors should also say if there are any convergence guarantees. The literature on recovery in SBMs, for example, gives concrete bounds on the recovery error (~ \sqrt{t} misclassifications, for example). Are such results possible in the general case?

Overall, I feel the paper needs a much more accessible exposition for a NeurIPS audience. But this said, the results do seem interesting if presented in the right way.

---

> ### Author Rebuttal · Authors · 2025-07-29
>
> We thank the reviewer for their thoughtful comments and for finding our work interesting. We address the main concerns raised by the reviewer below.
>
> **[W1&2] We will highlight relevance and improve accessibility to ML community.** We agree with the reviewer that our contributions to the ML community, i.e., **a complete characterization of MPNN learnability**, should be further highlighted to make the paper more appealing and accessible. We propose the following changes to this end:
>
> 1. **We will highlight the broader context to our characterization of MPNN learnability**. In the introduction and Section 2.6, we will clearly position our work within the context of commonly used random graph models for MPNN analysis. Existing work investigates specific parameterizations of sparse random graphs [1], or fixed graph size distributions [3], or restricted classes of MPNNs on dense random graphs [4], whereas **our work gives a general and complete characterization of random graph models on which fully expressive MPNNs can be learnt**.
>
> 2. **We will discuss that Theorem 3.6 generalizes and contextualizes existing machine learning results** like [1], [2] and [4]. In particular, we will highlight that Theorem 3.6 separates the following widely investigated random graph models in terms of MPNN learnability:
> + **Learnable.** Sparse SBM, preferential attachment models, inhomogeneous random graphs, and RCM parametrizations satisfy color convergence. Our results provide consistency guarantees for these random graph families, significantly generalizing previous results [1].
> + **Not Learnable.** Dense random graphs (like SBM and Erdős-Rényi) with growing average degree are incompatible with MPNN learning. This result expands on the impossibility of size generalization [2], previously only identified for specific random graphs, and adds a broader context to results on limit behavior of MPNNs [4] for such random graphs.
>
> 3. We will expand Section 3.1 with a natural example (such as dense Erdős–Rényi or SBM) on how **MPNN learning fails on random graphs with growing expected degree**.
>
>
> **[W3] Convergence Rates.** The generality of our results critically relies on taking a non-parametric approach (note that $\mu$ in Definition 4.1 is an arbitrary distribution with infinite domain). However, this approach precludes using standard parametric techniques, such as those for community recovery in SBMs.
> **The RCM can encode diverse limit behaviors**, e.g. sparse SBM if $\mu$ is multivariate Poisson, bipartite configuration model if there are 2 types and $\mu(s, A) = 0$ for $s\in A$, and so on. **This flexibility naturally leads to different convergence phenomena, which can exhibit considerable complexity,** for example when the distribution $\mu$ has heavy tails. Analysing convergence rates in general and for specific parameterizations constitutes interesting future work.
>
> **[Q1]** Regarding Theorem 3.13, **involution invariance and finite expected degree are fundamental to local convergence theory [5]. We extend their validity to the weaker setting of color convergence.** Furthermore, Corollary 3.14 provides a convenient criterion for color convergence. We will add a remark highlighting this fact.
>
> [1] Baranwal, A., Fountoulakis, K., & Jagannath, A. (2023). Optimality of message-passing architectures for sparse graphs. *Advances in Neural Information Processing Systems, 36.*
>
> [2] Yehudai, G., Fetaya, E., Meirom, E., Chechik, G., & Maron, H. (2021). From local structures to size generalization in graph neural networks. *38th International Conference on Machine Learning.*
>
> [3] Maskey, S., Levie, R., Lee, Y., & Kutyniok, G. (2022). Generalization analysis of message passing neural networks on large random graphs. *Advances in neural information processing systems, 35.*
>
> [4] Adam-Day, S., Benedikt, M., Ceylan, I., & Finkelshtein, B. (2024). Almost surely asymptotically constant graph neural networks. *Advances in Neural Information Processing Systems, 37.*
>
> [5] Aldous, D., & Lyons, R. (2007). Processes on unimodular random networks. *Electronic Journal of Probability, 12.*

---

### Note · Authors · 2025-08-15

We thank all the reviewers for their constructive feedback. In the following, we summarize the main concerns they raised and our proposed resolutions.

1. **Relevance to ML** (Reviewer 1zSd):

Our results generalize and contextualise multiple existing results on GNN learnability and limit behavior of GNNs. We will emphasize the relation of our results to existing literature more concretely (see response to Reviewer 1zSd, W1&2).

2. **Empirical results** (Reviewers X2fm and wyHK):

Our results provide a non-parametric characterization for learnability. Any experiments would require concrete parametrization choices, which might not be representative of our more general results. In restricted scenarios, we will highlight existing empirical evidence from ML literature for our results (see response to Reviewer X2fm, W3).

3. **Convergence rates** (Reviewers 1zSd,  X2fm and HKEY):

Even for simple parametrizations the RCM exhibits very diverse limit behavior (e.g. sparse SBM if $\mu$ is multivariate Poisson, bipartite configuration mode if there are 2 types and $\mu(s, A) = 0$ for $s\in A$, etc.). Analyzing convergence rates under different parametric or non-parametric assumptions constitutes interesting future work (see response to Reviewer 1zSd, W3).

---

### Decision · Program_Chairs · 2025-09-17

**Decision:**

Accept (poster)

**Comment:**

The paper introduces a variant of Benjamini-Schramm convergence of sparse graphs, and relates it to node classification behavior of message passing graph neural networks.

This is an interesting idea and the paper is well-written. There are some minor weaknesses; in particular, there is no experimental valuation, which is not a strict requirement for NeurIPS, but as mentioned in the reports, this paper may have benefited from including it. Overall, however, this is interesting work and the reviews and scores support publication.